# A C_2_H_2_ Zinc Finger Protein PlCZF1 Is Necessary for Oospore Development and Virulence in *Peronophythora litchii*

**DOI:** 10.3390/ijms23052733

**Published:** 2022-03-01

**Authors:** Honghui Zhu, Junjian Situ, Tianfang Guan, Ziyuan Dou, Guanghui Kong, Zide Jiang, Pinggen Xi

**Affiliations:** Department of Plant Pathology/Guangdong Province Key Laboratory of Microbial Signals and Disease Control, South China Agricultural University, Guangzhou 510642, China; hui57464840@163.com (H.Z.); junjian.st@hotmail.com (J.S.); guantianfang@163.com (T.G.); dzyxfrsc@163.com (Z.D.); gkong@scau.edu.cn (G.K.); zdjiang@scau.edu.cn (Z.J.)

**Keywords:** *Peronophythora litchii*, oospore, C_2_H_2_ zinc finger protein, transcription factor

## Abstract

C_2_H_2_ zinc finger is one of the most common motifs found in the transcription factors (TFs) in eukaryotes organisms, which have a broad range of functions, such as regulation of growth and development, stress tolerance and pathogenicity. Here, *PlCZF1* was identified to encode a C_2_H_2_ zinc finger in the litchi downy blight pathogen *Peronophythora litchii*. *PlCZF1* is conserved in *P. litchii* and *Phytophthora* species. In *P. litchii*, *PlCZF1* is highly expressed in sexual developmental and early infection stages. We generated *Δplczf1* mutants using the CRISPR/Cas9 method. Compared with the wild type, the *Δplczf1* mutants showed no significant difference in vegetative growth and asexual reproduction, but were defective in oospore development and virulence. Further experiments revealed that the transcription of *PlM90*, *PlLLP* and three laccase encoding genes were down-regulated in the *Δplczf1* mutant. Our results demonstrated that *PlCZF1* is a vital regulator for sexual development and pathogenesis in *P. litchii.*

## 1. Introduction

To regulate a wide range of physiological processes, organisms utilize transcription factors (TFs) to activate or repress gene transcription by binding to specific DNA sites. In eukaryotic organisms, TFs can be divided into different classes based on the DNA-binding domain types, including basic region leucine zipper (bZIP), MADS-box, MYB, helix-loop-helix, homeobox and zinc fingers [1,2,3,4,5,6,7]. C_2_H_2_ zinc fingers are the most common DNA-binding motifs found in eukaryotic transcription factors, and have also been identified in prokaryotes [8]. They are relatively small protein motifs which contain multiple finger-like protrusions that make tandem contacts with their target molecules. The classical C_2_H_2_ zinc fingers contain a short beta hairpin and an alpha helix, where a single zinc atom is held in place by Cysteine_2_Histidine_2_ (C_2_H_2_) residues in a tetrahedral array [9]. They can be divided into three groups based on the number and pattern of fingers: triple-C_2_H_2_ (binds single ligand), multiple-adjacent-C_2_H_2_ (binds multiple ligands), and separated paired-C_2_H_2_ [10]. C_2_H_2_ zinc fingers have a broad range of functions, such as regulation of growth and development, stress tolerance and pathogenesis in eukaryotic and prokaryotic organisms [11,12].

Recently, there have been some advances in the study of C_2_H_2_ zinc fingers in fungi/oomycetes. For example, the *Aspergillus nidulans* gene *aslA* encoding a C_2_H_2_ zinc finger is related to K^+^ stress resistance, vacuolar morphology and vacuolar transporters [13]. RS_CRZ1, a C_2_H_2_ zinc fingers from *Rhizoctonia solani*, was reported to contribute to the pathogen colonization by the host-induced gene silencing (HIGS) approach [14]. *PsCZF1* is required for growth, development and pathogenesis in oomycete pathogen *Phytophthora sojae* [15]. However, the functions of C_2_H_2_ zinc finger in fungi/oomycetes were still largely unknown.

Oomycetes include many phytopathogens that infect a wide range of ornamentally and agriculturally important plants. Although phytopathogenic oomycetes exhibit a fungus-like filamentous growth morphology, they belong to *Stramenopila* kingdom, which are distant relative to fungi in evolution [16]. The litchi downy blight pathogen *Peronophythora litchii* (also named *Phytophthora litchii*) is one of the notorious phytopathogenic oomycetes, which distributes throughout most litchi-growing regions and causes fruit reductions each year [17,18]. This disease occurs during the growth and development of litchi trees, and is able to infect the tender leaves, panicles and fruits.

The thick-walled oospores, generated by oomycete sexual reproduction, play key roles in many of disease process. Oospores are usually capable of long-term survival away from the host plant environment, such as soil or plant debris. They germinate at the start of a growing season as the source of primary inoculum for disease epidemics. According to their mating style, oomycetes can be divided into two different types: heterothallism (oospore generation with A1 and A2 mating types); or homothallism (oospore generation without crossing of opposite mating strains) [19]. *P. litchii* are the important homothallic organism causing serious plant disease.

Although the available genome and recent established CRISPR-mediated gene knockout system strengthened the functional genomic research in *P. litchii* [20,21,22,23,24], the pathogenesis of this pathogen is still largely unknown. In this study, we report a gene *PlCZF1* encoding C_2_H_2_ zinc finger in *P. litchii* that is highly induced in oospores and the early infection phase. CRISPR-mediated gene knockout demonstrated that it plays key roles in the development of oospore and pathogenesis on litchi. Subcellular localization assays showed that PlCZF1 mainly distributes in nuclei of mycelia and sporangia. The transcription level of *PlM90*, *PlLLP* and laccase encoding genes are also regulated by PlCZF1. Our study provides insights into the C_2_H_2_ zinc finger TF functions during oospore formation and the pathogenesis of oomycetes.

## 2. Results

### 2.1. PlCZF1 Is a Member of the C_2_H_2_ Zinc Finger Protein Family

Based on the sequence of C_2_H_2_ zinc finger encoding gene *PsCZF1* in *p**. sojae* (GenBank ID: EU912575.1) [15] and the available genome data of *P. litchii* [20], we identified a gene the most identical to the *PsCZF1* in sequence and named it *Peronophythora litchii* C_2_H_2_ zinc finger 1 (PlCZF1). The *PlCZF1* gene is 1428 bp in length with no intron, and encodes a protein of 476 amino acids (aa), which is 84% identity with PsCZF1 in protein level. The structural analysis showed that PlCZF1 contains a bromodomain and extraterminal (BET) domain followed by four Znf-C_2_H_2_ domains (Figure 1A). Phylogenetic analysis showed that PlCZF1 was in a clade including its orthologs of the *Phytophthora* and the downy mildew species (Figure 1B). The clades of the orthologs of downy mildew, *Pythium insidiosum* and *Saprolegnia parasitica,* were outside, but closest to that of *Phytophthora* species, while those of *Saccharomyces cerevisiae* and *Aspergillus fumigatus* exhibited the farthest distance from the oomycete cluster (Figure 1B). These results suggest that the PlCZF1 is highly conserved in *Phytophthora* species.

### 2.2. PlCZF1 Localizes to Nucleus and Induced in Oospores and the Early Infection Phase

To investigate the temporal and spatial pattern of PlCZF1 expression, we made a construct of PlCZF1 fused to the RFP (Figure 2A) and introduced it into *P*. *litchii* using the PEG-mediated protoplast transformation method. Hyphae and sporangium of the resulting transformants expressing the full-length PlCZF1 fused to RFP were examined by fluorescence microscopy. The results showed that RFP fluorescence was localized in the nucleus of mycelia and sporangia (Figure 2B).

We also examined the expression patterns of *PlCZF1* at different development and host infection stages using qRT-PCR. Comparing with that of mycelia stage, we found the expression levels of *PlCZF1* were highly induced in oospores and the early infection phase (Figure 3), suggesting that the *PlCZF1* play a role in oospore development and pathogenesis.

### 2.3. Generation of PlCZF1 Knockout Mutants by CRISPR/Cas9

To investigate the biological functions of *PlCZF1*, we generated *PlCZF1* deletion mutants using the CRISPR-mediated gene knockout method. Two single guide (sg) RNAs were designed to disrupt the *PlCZF1* coding region (Figure 4A) by a previously described protocol [25]. A total of 61 G418-resistant transformants were isolated, and used for further PCR analysis. Three of the transformants (T1, T11 and T16) generated smaller bands than wild type (WT) strain and non-knockout mutant (Control, CK), indicating the deletion event (Figure 4B). To further ensure the knockout of *PlCZF1* in *P. litchii*, sanger sequencing was performed, and the results confirmed that *PlCZF1* were deleted (Figure 4C). The expression levels of *PlCZF1* in the knockout mutants were also tested by qRT-PCR, and the results showed that *PlCZF1* was completely absent in T1, T11 and T16 (Figure 4D).

### 2.4. PlCZF1 Does Not Affect Mycelium Growth, Sporangium Production and Zoospore Release

To explore the biological functions of *PlCZF1* in *P. litchii* development, we first measured mycelium growth rate of the three knockout mutants, as well as WT and CK strains. As shown in Figure 5A,B, the mycelium growth rate of the three *Δ**plczf1* knockout mutants (11.5~12.36 mm/d) were the same as WT (12.26 mm/d) and CK (11.86 mm/d) strains. Then, we assessed the production of *Δ**plczf1* sporangia. The average numbers of T1, T11 and T16 of the three *Δ**plczf1* per μL sporangium suspension were 18.33, 19.33 and 20 respectively, no different with WT (17.66) and CK (16.33) strains (Figure 5C). We further investigated zoospore release and cyst germination rate of the *Δ**plczf1*, and the results showed that zoospore release and cyst germination rate were not affected by the deletion of *PlCZF1* (Figure 5D,E).

### 2.5. PlCZF1 Is Essential for Oospore Development

Since *PlCZF**1* did not affect the vegetative growth and asexual sporulation of *P. litchii*, we compared the oospore development of the *Δ**plczf1* mutants and controls. As shown in Figure 6A, WT and CK strains showed normal oospore exhibiting thinner oospore walls and filling with ooplast after 10 days of growth on CJA medium. In contrast, the proportion of abnormal oospores that the ooplast was completely or partially void in the *Δ**plczf1* is more than 95%, significantly higher than that of WT and CK strains (Figure 6B).

The RNA binding protein M90 were previous reported to positive regulation of oospore production in oomycetes [26,27]. Loss of the loricrin-like protein (LLP)-encoded gene *PiLLP* led to defects in the formation of normal cytoplasm in oogonia in *P. infestans* [28]. In order to determine whether these three genes were influenced by *PlCZF1*, the transcriptional level of *PlM90*, *PlLLP* were evaluated in the *Δ**plczf1* mutants. The qRT-PCR results showed that the transcript levels of *PlM90* and *PlLLP* in *Δ**plczf1* mutants were down-regulated compared with those in WT (Figure 6C), indicating that *PlCZF1* might affect oospore development through regulating the transcription of *Pl**M90* and *PlLLP*.

### 2.6. PlCZF1 Is Required for Pathogenicity

To determine the contribution of PlCZF1 on *P. litchii* virulence, we performed an inoculation assay. The WT and CK strains, as well as *Δ**plczf1* mutants, were inoculated on litchi leaves. The diameter of lesions caused by the pathogen was measured at 48 h post-inoculation. Our results showed that the lesions caused by the *Δ**plczf1* mutants were obviously smaller than those caused by wild type and CK strains (Figure 7A,B), suggesting that *PlCZF1* plays an important role in *P. litchii* virulence. 

### 2.7. PlCZF1 Regulate Extracellular Laccase Activity 

Laccase and peroxidase secreted by fungi/oomycete have been reported to contribute to pathogenicity by detoxifying the increase of reactive oxygen species (ROS) induced by the plant defense response [29,30], and thus we further assessed these extracellular enzymes activities in *Δ**plczf1*. Congo Red (CR) degradation assay was first tested for examining the peroxidase activity. As shown in Figure 8A (Upper panel) and B, diameters of the halo caused by the three *Δ**plczf1* were comparable to that of WT and CK strains. Subsequently, we investigated laccase activity based on oxidation assay of ABTS [2,20-azino-bis (3-ethylbenzothiazoline-6 sulfonic acid)]. Compared with WT and CK strains, all the *Δ**plczf1* showed significantly decreased accumulation of ABTS, which was indicated by dark purple staining around the mycelial mat (Figure 8A, Lower panel and C). These results revealed that *PlCZF1* may regulate extracellular laccase activity to ensure successful host infection.

Additionally, we also examined the transcription of putative laccase encoding genes in *Δ**plczf1* according to previous studies [21]. The results showed that the transcript levels of two predicted laccase genes were not obviously changed in three *Δ**plczf1* compared with WT and CK strains (Figure 8D). In contrast, three laccase encoding genes (*Pl_103272*, *Pl_10**4952* and *Pl_106181*) out of the five predicted laccase genes showed a significant decrease (47–75%) in transcript level in *Δ**plczf1* mutant compared to that of WT and CK strains (Figure 8D). Knockout of *PlCZF1* affecting the transcript level of a predicted laccase gene, we speculate that *PlCZF1* plays a regulatory role in *P. litchii* pathogenesis through direct or indirect regulation on the transcription of laccase-encoding genes.

## 3. Discussion

The C_2_H_2_ zinc finger are widespread in fungal/oomycete pathogens, being involved in development, metabolism and virulence [31,32]. In this paper, we report a putative C_2_H_2_ zinc finger PlCZF1, from *P. litchii*. CZF1 protein was conserved in *Phytophthora*, and the downy mildew species but was distinct from fungi. Transcriptional profile analysis showed that *PlCZF1* was significantly up-regulated in oospore and the early infection stage. Subsequently, functional studies demonstrated that *PlCZF1* was important for oospore development and virulence on host litchi.

As a homothallic oomycete, *P. litchii* increases genetic fitness and diversity, and provides a durable agent for spreading disease by sexual sporulation; while asexual reproduction generates sporangia and zoospores that are the main agents for the rapid dispersal of disease [21]. To investigate the mechanism of sporulation and infection in oomycete, we used PlCZF1 as an example, and made *Δ**plczf1* mutants by CRISPR-mediated gene knockout. In previous studies, C_2_H_2_ zinc finger have been reported in fungi that are involved in regulating growth and development correct responses to a variety of external and internal stresses. For example, the recent study on two *Fusarium graminearum* C_2_H_2_ zinc finger Fg01341 and Fg01350 showed that Fg01341 was associated with sexual reproduction and virulence, and Fg01350 regulated hyphal growth, sexual production, virulence and deoxynivalenol production [33]. Silencing of C_2_H_2_ zinc finger gene *PsCZF1* in *P. sojae* led to reduced mycelia growth, production of oospores and swimming zoospores, and also impair pathogen virulence [15]. Here, we did not observe the effects of *PlCZF1* on mycelia growth and asexual reproduction of *P. litchii*. However, we found that *PlCZF1* is required for oospore development and virulence. Combining the results of PsCZF1 and our study, we found a common critical role of the CZF1 at oospore development of oomycetes. As a fruit tree pathogen, *P. litchii* has a narrow host range and invades litchi during flower and fruit formation period (usually March to August). The rest of the time, oospores produced by *P. litchii* survive in soil or plant debris for resisting harsh environmental conditions and germinate as the secondary source of infection. Thus, understanding the oosporogenesis of *P. litchii* will facilitate development of effective control strategies of litchi downy blight.

Intriguingly, previous study has revealed that the Puf RNA-binding protein PlM90 is essential for oospore formation in *P. litchii* [26]. Our results demonstrated that the transcript levels of *PlM90* were down-regulated in *Δ**plczf1* mutant, which hinted the relationship between *PlCZF1* and *PlM90*. Future studies will be done to reveal whether *PlCZF1* target the transcription control region of *PlM90* or other genes involved in oospore development.

The virulence of *P*. *litchii* was severely impaired when *PlCZF1* was deleted. Knockout of *PlCZF1* did not affect growth rate and asexual reproduction, and thus we speculated that it was associated with infection process by regulating the secretion of detoxification enzymes, such as peroxidase and laccase. The extracellular enzymatic activity test and subsequent qRT-PCR analysis confirmed the regulation of laccase-encoding genes by *PlCZF1*. Additionally, whether other secretory proteins associated with virulence were also controlled by *PlCZF1* requires further studies.

## 4. Materials and Methods

### 4.1. Phylogenetic Analysis

Genome sequence of *P. litchii* used in this study were retrieved from NCBI (BioProject ID: PRJNA290406). Sequence alignment (Muscle algorithm) and phylogenetic tree were constructed with the neighbour-Joining algorithm with 1000 bootstrap replications in the MEGA 7.0 program (http://megasoftware.net, accessed on 2 May 2021). The conserved domains were predicted using SMART (http://smart.embl-heidelberg.de, accessed on 5 April 2021).

### 4.2. Microbial Strains and Culture Conditions

*P. litchii* strain SHS3 was isolated from Guangzhou, Guangdong Province, China and storage in mycological laboratory of department of plant pathology, South China Agricultural University. The strain was cultured on carrot juice agar (CJA) medium (juice from 200 g carrot topped up to 1 L, 15 g agar/L for solid medium) at 25 °C in the dark. *Escherichia coli* DH5α was cultured at 37 °C in Luria Bertani (LB) medium and used for the cloning and propagation of recombinant plasmids.

### 4.3. Plasmid Construction and CRISPR-Mediated Gene Knockout

All the primers used in this study are listed in Appendix A. The PCR fragments were amplified by Phanta Max Super-Fidelity DNA Polymerase (Vazyme, Nanjing, China). To construct the PlCZF1-RFP fusion expression vector, the full length of *PlCZF1* was linked to the linearized pTORmRFP4, which was digested by ClaI and BsiWI (New England Biolabs, Hitchin, UK). The vectors pYF2.3G-RibosgRNA used for knockout of *PlCZF1* by CRISPR/Cas9 were generated as described previously [25]. Transformation of the *P. litchii* strain SHS3 was performed as described previously [22].

### 4.4. Fluorescence Microscopy

To visualize the subcellular localization of PlCZF1 in *P. litchii* mycelia and sporangia of the PlCZF1-RFP fusion expression, strains were stained with the blue-fluorescent nucleic acid stain 4′,6-diamidino-2-phenylindole (DAPI), dilactate (Invitrogen, Carlsbad, CA, USA), and then viewed using an Olympus BX53 microscope (Olympus, Tokyo, Japan). The fluorescence was detected with fluorescence filter BP340-390 and BP530-550 for DAPI and RFP, respectively. Three random fields at 60× magnification from each sample were selected for observed.

### 4.5. RNA Extraction and Gene Expression Analysis

Mycelia, sporangia, zoospores, cysts, germinating cysts, oospores and litchi leaves infected with mycelial mats of *P. litchii* were harvested at the indicated time points, and RNA was extracted using All-In-One DNA/RNA Mini-preps Kit (Bio Basic, Markham, ON, Canada) according to the recommended protocol. All cDNAs were synthesized from total RNA by PrimeScript RT Master Mix (Takara, Shiga, Japan). qRT-PCR was performed in 20 μL reactions that included 20 ng cDNA, 0.4 μΜ gene-specific primer, 10 μL SYBR Premix ExTaq II (Takara) and 6.4 μL ddH_2_O. The qRT-PCR were performed on qTOWER3 Real-Time PCR thermal cyclers (Analytik Jena, Jena, Germany) under the following conditions: 95 °C for 2 min, 40 cycles at 95 °C for 30 s, and 60 °C for 30 s to calculate cycle threshold values, followed by a dissociation program of 95 °C for 15 s, 60 °C for 1 min, and 95 °C for 15 s to obtain melt curves. The relative expression values were determined using *PlActin* as reference gene and calculated with the formula 2^−ΔΔCt^.

### 4.6. Pathogenicity Test

For pathogenicity assays, mycelial mats were inoculated on the tender leaves of litchi cultivar ‘Feizixiao’ collected from the litchi orchard in South China Agricultural University, Guangzhou, Guangdong Province. The mycelial mat of each strain was inoculated on the center of the leaf, and maintained at 80% humidity in 12 h light/12 h darkness at 25 °C. Each strain was tested on no fewer than 30 leaves. The lesion diameter was observed and measured at 48 h post-inoculation. The experiments were repeated at least three times.

## 5. Conclusions

In conclusion, we characterized the functions of PlCZF1 in *P. litchii* with CRISPR/Cas9-mediated genome editing method. Our data revealed that PlCZF1 is conserved in oomycetes, and plays a key role in oospore development, virulence and regulation of laccase activity and transcription of *PlM90* and *PlLLP* in *P. litchii*. The results of this study suggested that PlCZF1 could be a target for developing novel strategies for disease management. Additionally, the results in this study may inspire further regulation mechanism studies. In particular, it would be interesting to study the downstream genes regulated by PlCZF1 or interaction proteins of PlCZF1, which will provide a new way to decipher the pathogenesis of *P. litchii*.

## Figures and Tables

**Figure 1 ijms-23-02733-f001:**
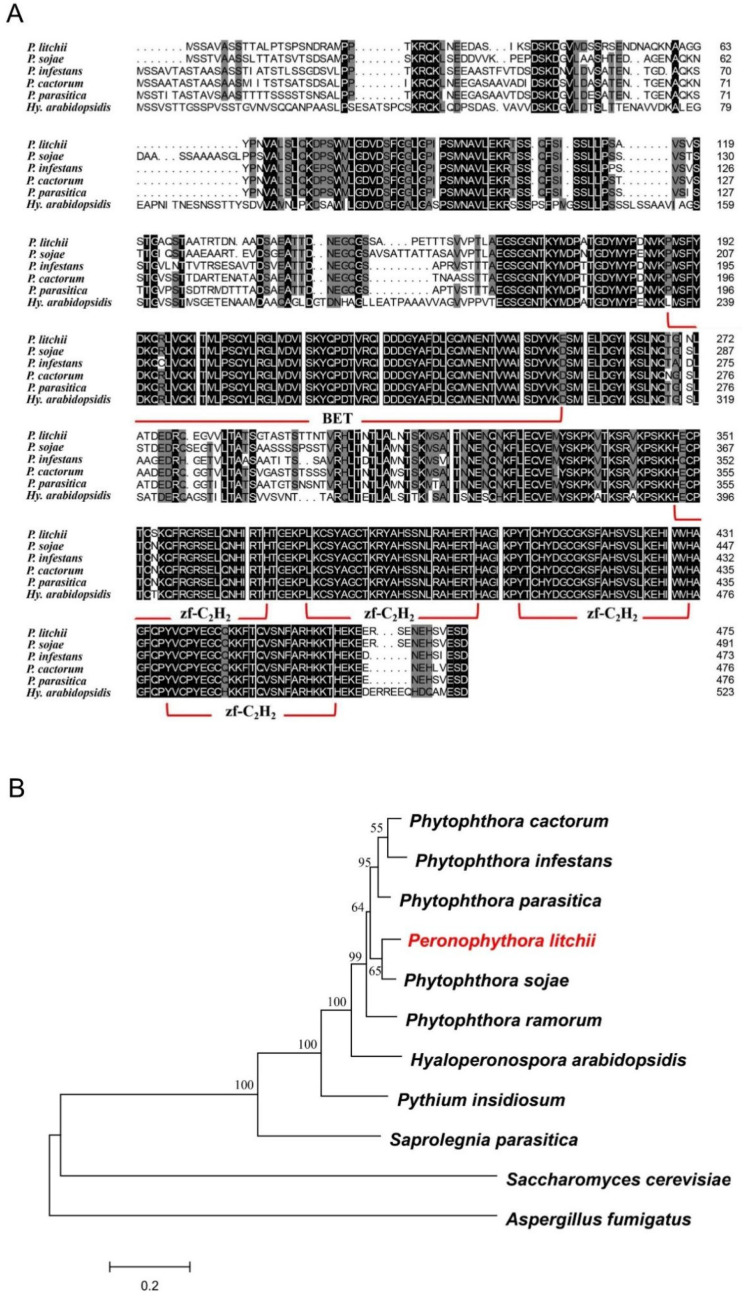
Domain arrangement and phylogenetic analysis with PlCZF1 protein and its orthologs. (**A**) Protein sequence alignment of PlCZF1 and its orthologs from *Phytophthora infestans*, *Phytophthora sojae*, *Phytophthora cactorum*, *Phytophthora parasitica* and *Hyaloperonospora arabidopsidis*. Columns with identical and similar amino acid sequences were colored with black and gray, respectively. The regions of BET and Znf-C_2_H_2_ were indicated. (**B**) Phylogenetic analysis of CZF1 proteins of *P. litchii* and other oomycetes and fungi. Phylogenetic dendrograms were constructed by MEGA 7.0, with the Neighbour-Joining method using 1000 bootstrap replications.

**Figure 2 ijms-23-02733-f002:**
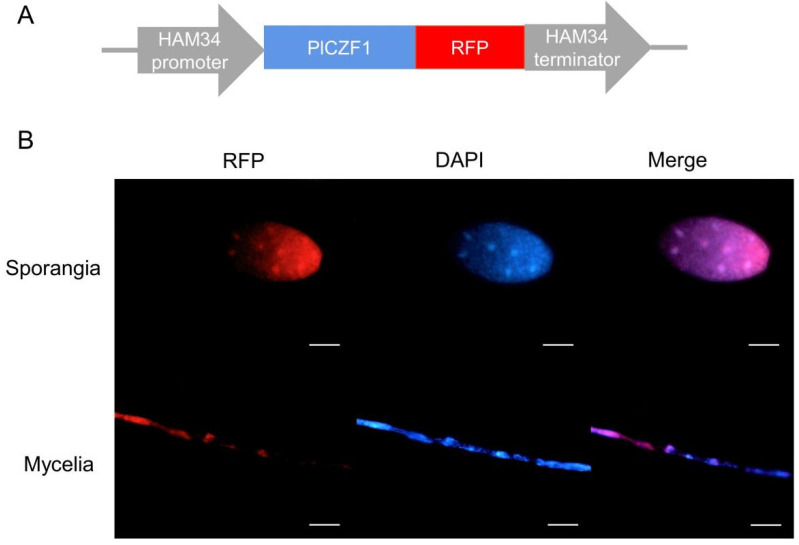
Subcellular localization of PlCZF1. (**A**) The illustration of the construct produced for PlCZF1 localization. (**B**) The hypha and sporangia of the fluorescent transformants of *P. litchii* were observed under a fluorescence microscope during red fluorescence (RFP) and DAPI (blue) staining. The scale of the sporangium stage is 5 μm, and the scale of the hyphae stage is 10 μm.

**Figure 3 ijms-23-02733-f003:**
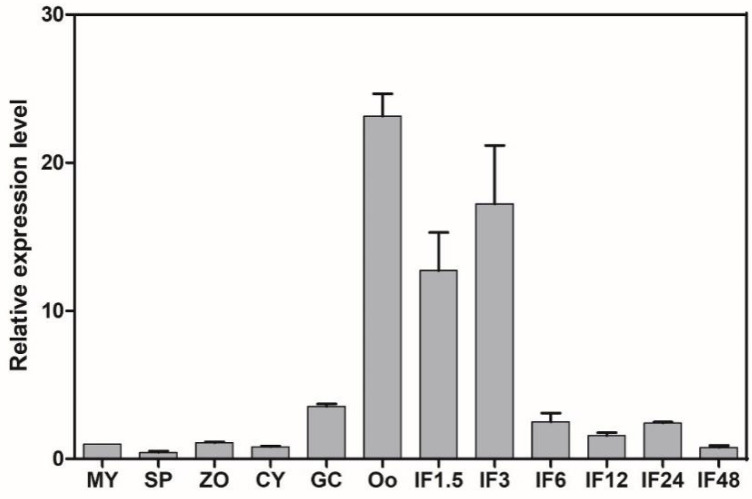
Expression profile *PlCZF1*. Relative expression levels of *PlCZF1* were determined by qRT-PCR with total RNA extracted from mycelia (MY), sporangia (SP), zoospores (ZO), cysts (CY), germinating cysts (GC) and oospores (Oo), and litchi leaves infected by *P. litchii* mycelial mats after 1.5, 3, 6, 12, 24 and 48 h (IF1.5 h to IF48 h). The constitutive expression of *PlActin* was used as reference gene. Expression levels were normalized using the MY values as ‘1’. Data represent means ± SD from three independent biological repeats.

**Figure 4 ijms-23-02733-f004:**
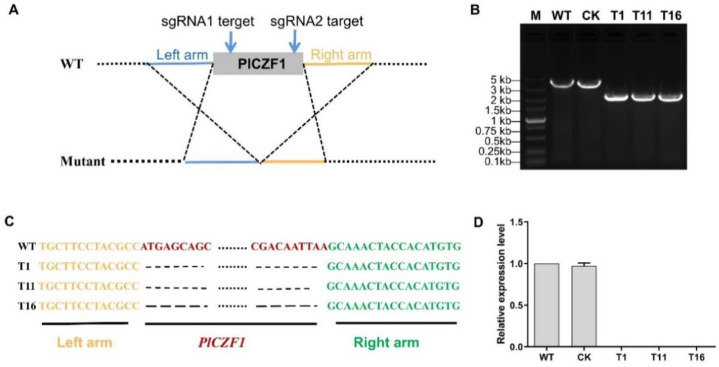
CRISPR/Cas9-mediated knockout of *PlCZF1*. (**A**) Two single guide RNAs targeting the two sides of the PlCZF1 coding region were designed to disrupt *PlCZF1*. (**B**,**C**) Three independent deletion mutants were identified by genomic PCR and confirmed by sequencing. (**D**) Relative expression levels of *PlCZF1* were determined by qRT-PCR in WT, CK, T1, T11 and T16. The constitutive expression of *PlActin* was used as a reference gene. Expression levels were normalized using the WT values as ‘1’. Data represent means ± SD from three independent biological repeats.

**Figure 5 ijms-23-02733-f005:**
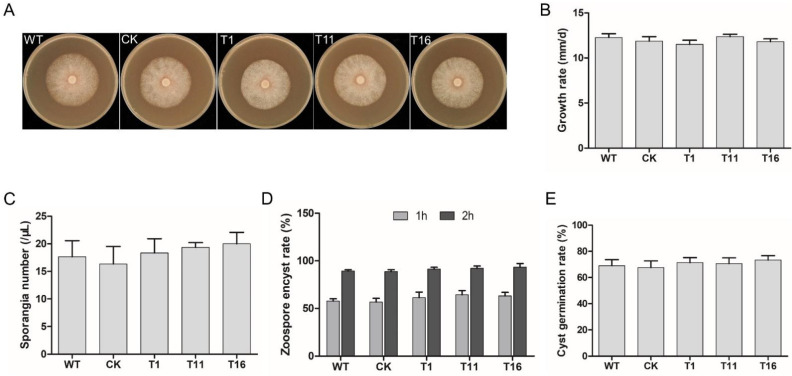
Growth rate and asexual reproduction analysis of *Δ**plczf1* mutants. (**A**,**B**) Colony growth rate of WT, CK and the three *Δ**plczf1* mutants. Photographs were taken at 7 days. This experiment was repeated three times independently, and for each repeat with three biological replications. (**C**) Mean sporangium number in 1 μL sporangium suspension. (**D**) Mean zoospore encyst rate. Sporangia were harvested by flooding the mycelia with the same sterile distilled water. (**E**) Mean cyst germination rate. For (**C**,**D**), the initial concentration of cysts, sporangia and zoospores was adjusted to be the same for each strain. The error bar presented the standard error. The data were statistically analyzed with SPSS (version 19.0) with Duncan’s Multiple Range Test method.

**Figure 6 ijms-23-02733-f006:**
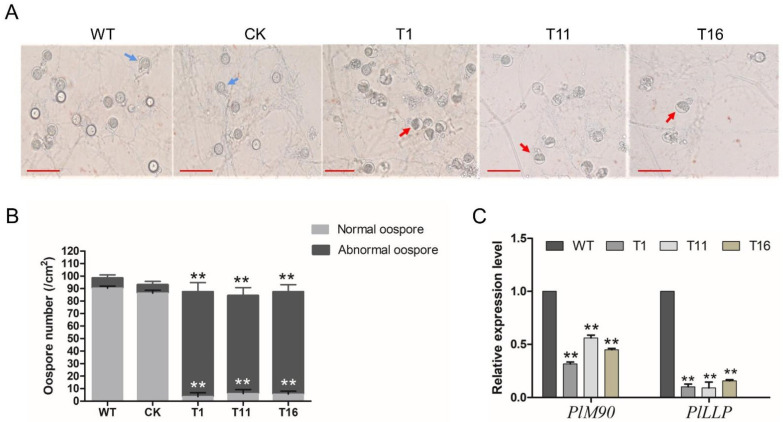
Deletion of *PlCZF1* produced many aborted oospores. (**A**) Morphology of oospores produced in 17 days old cultures of WT, CK and Δplczf1 mutants. Blue arrows indicate the normal oospore and red arrows indicate the abnormal oospore. The bars indicate 20 μm. (**B**) Amount of normal and aborted oospores produced by WT strain, control strain and the aborted mutants. Data represent means ± SD from three independent biological repeats, and asterisks indicate significant differences (** *p* < 0.01 compared with WT, Dunnett’s test). (**C**) The relative expression level of *PlM90*, *PlLLP* in *Δplczf1* mutants. The constitutive expression of *PlActin* was used as reference gene. Data represent means ± SD from three independent biological repeats, and asterisks denote significant differences from the control group (Student’s *t* test: ** *p* < 0.01).

**Figure 7 ijms-23-02733-f007:**
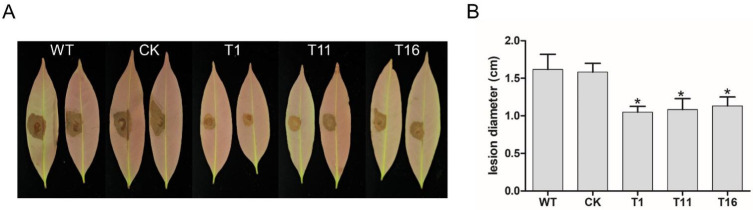
Pathogenicity test of *Peronophythora litchii*
*Δ**plczf1* mutants, WT and CK strains. (**A**) Mycelial mats were inoculated on tender litchi leaves (*n* = 30 for each strain) for 48 h at 25 °C in the dark. Images are two representatives for each instance. (**B**) Quantification of pathogenicity by disease severity values. Asterisks denote a significant difference between *Δ**plczf1* mutants and two controls (Dunnett’s test, * *p* < 0.05). This experiment was repeated three times independently, and each repeat with three biological replications.

**Figure 8 ijms-23-02733-f008:**
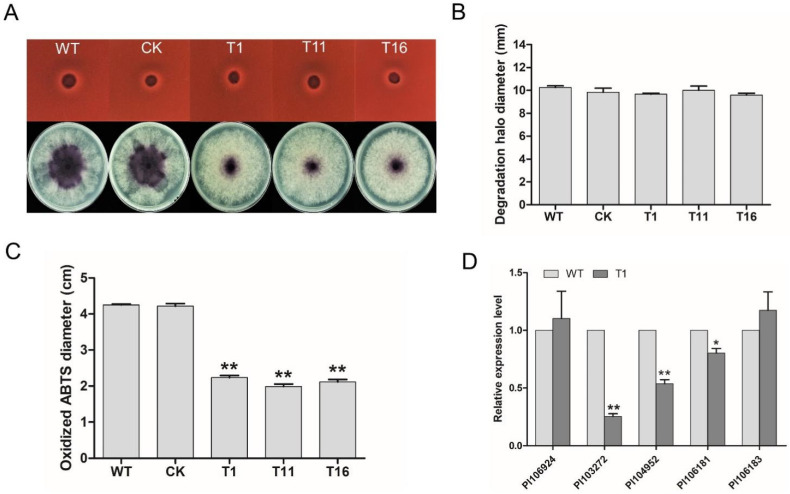
Detection of extracellular peroxidase and laccase activity in *PlCZF1* deletion mutants. (**A**) Peroxidase and laccase activity assay. For peroxidase activity assay, mycelial mats of WT, CK and three *PlCZF1* deletion mutants were inoculated onto solid Plich medium containing Congo red at a final concentration of 500 g/mL for 1 day; For laccase activity assay, mycelial mats of the aforementioned strains were inoculated on lima bean agar (LBA) media containing 0.2 mM ABTS for 10 days. (**B**) The discoloration halo diameters were measured at 1 day post-inoculation. (**C**) The diameters of oxidized ABTS (dark purple) were measured at 10 days post-inoculation. Different letters bar charts represent a significant difference (*p* < 0.05) based on Duncan’s multiple range test method. These experiments contained three independent biological repeats, each of which contained three replicas. (**D**) The transcript levels of putative peroxidase-encoding genes and laccase-encoding genes in T1 and WT strain. The constitutive expression of *PlActin* was used as a reference gene. Data represent means ± SD from three independent biological repeats, and asterisks denote significant differences from the control group (Student’s *t*-test: * *p* < 0.05; ** *p* < 0.01).

## Data Availability

Not applicable.

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
