# Peer review of "A C2H2 Zinc Finger Protein PlCZF1 Is Necessary for Oospore Development and Virulence in Peronophythora litchii"

_ijms, 2022, doi:10.3390/ijms23052733_

Round 1
Reviewer 1 Report
In this manuscript, authors characterized the function of PlCZF1, a C2H2 transcription factor, found in oomycete Peronophythora litchii, which causes litchi downy blight. Phylogenetic analyses showed that CZF1 proteins are highly conserved among Phytophthora species. Microscopic analysis using PEG-mediated protoplast transformation of RFP-tagged PlCZF1 showed nuclear localization of PlCZF1. qRT-PCR analysis revealed that PlCZF1 highly expresses in oospores and at early infection stage, but not so much in mycelia, sporangia, zoospores, and cysts. To examine the function of PlCZF1, authors performed knockout mutant of PlCZF1 by CRISPR/Cas9. The deletion of PlCZF1 was confirmed by genotyping. plczf1 mutant showed impaired oocyte development. The expressions of M90 and LLP, both of which are associated with oomycete sexual development, were reduced in plczf1. plczf1 mutant showed reduced virulence on host litchi and reduced laccase activity.
Overall, this manuscript contains enough data with good quality, and I have only minor comments as listed below.
Minor comments
- Please explain how authors selected PlCZF1 for this research.
- Please give detailed information for the functions of M90 and LLP.
- Please add an illustration of the construct produced for PlCZF1 localization analysis to show which promoter was used and which terminus (N or C) RFP was fused.
- Is “Cytoplasmic localization” in Fig. 2 legend a mistake of “nuclear localization”?
- Please include qRT-PCR data to show that PlCZF1 expression is completely absent in plczf1.
- Please include the information of reference gene used for qRT-PCR analyses in each figure legend.
- While authors described that “the proportion of abnormal oospores in the plczf1 is more than 95%” (p5 L148~), it is not clear for readers who are not familiar with oomycete oospores which oospores are normal and which are abnormal from the pictures in Fig.6A. Please indicate the normal and abnormal oospores with arrows or arrowheads. Please also describe the morphological features of normal and abnormal oospores.
- Is the description in p7 L192~ regarding Fig. 8B a mistake of Fig. 8D?
- In Materials&Methods, all species names should be written in italic.
- Please give detailed information for fluorescence microscopy, such as the model of microscope, objective lens, and fluorescence filter.
Reviewer 2 Report
Dear Authors,
The manuscript presents comparative analysis of Peronophythora litchi wild type and the generated using the CRISPR/Cas9 method ∆plczf1mutants. The authors stated no differences in vegetative growth, and asexual reproduction, and significant differences in oospore development and virulence between wild and mutant strains. The methodologies of the experiments, applied molecular tools, and drawn conclusions are good and without objections I can recommend this manuscript for publication, however some minor correction should be introduced. I mean on:
L. 2
Instead of Peronophythora litchi please consider to use Phytophthora litchii which at present is recommended as valid name of the organism. The taxonomic status of the species and its belonging to Phythophtora genus confirms also fig 1 of this manuscript where Peronophythora litchi is located between Phytophthora species. See also: Goker M, Voglmayr H, Riethmuller A, Oberwinkler F. 2007. How do obligate parasites evolve? A multi-gene phylogenetic analysis of downy mildews. Fungal Genet Biol. 44:105–122
L. 23
Introduction should contain information on homothallic nature of the species. It seems to be important in context of oospore generation without crossing of opposite mating strains.
L. 91 and several times in other places of the manuscript
The authors use the term “mycelia” what is highly confusing as the Peronophythora litchi is not a member of Fungi kingdom, what Authors stressed in introduction. A less controversial statement seems to be “thallus”
L. 158 Fig. 6b
Student's t test is not appropriate statistical method for testing differences in number of oospore. It is not parametric trait.
L. 259
The P. litchii strain SHS3 is commonly cited in various papers, the authors should indicate the origin, source of the strain and its deposition place.
Reviewer 3 Report
The manuscript is well written and presented.
I have few minor comments.
The authors should mention in the Introduction that, C2H2 zinc fingers are relatively small protein motifs
or
C2H2 zinc fingers are relatively small protein motifs which contain multiple finger-like protrusions that make tandem contacts with their target molecule.
Moreover, the C2H2-type zinc finger proteins are a large family and probably authors should use C2H2 zinc finger proteins or be more specific using the C2H2 zinc finger (PsCZF1).
The major issue is that the authors need to specify the significance of this research compared with the first report of the putative C2H2 zinc finger transcription factor, PsCZF1, from Phytophthora sojae, reported by Wang et al. 2009.
Wang, Y., Dou, D., Wang, X., Li, A., Sheng, Y., Hua, C., ... & Wang, Y. (2009). The PsCZF1 gene encoding a C2H2 zinc finger protein is required for growth, development and pathogenesis in Phytophthora sojae. Microbial pathogenesis, 47(2), 78-86.
